# Technological Developments of Mobility in Smart Cities. An Economic Approach

**Javier Parra-Domínguez ***[ID], **Jorge Herrera Santos, Sergio Márquez-Sánchez** [ID]**, Alfonso González-Briones** [ID] **and Fernando De la Prieta** [ID]

BISITE Research Group, University of Salamanca, Edificio Multiusos I+D+i, 37007 Salamanca, Spain; jorgehsmp@usal.es (J.H.S.); smarquez@usal.es (S.M.-S.); alfonsogb@usal.es (A.G.-B.); fer@usal.es (F.D.l.P.)
* Correspondence: javierparra@usal.es

**Abstract:** This article introduces the concern that exists in the wider economic world concerning the developments carried out in Smart Cities. The various studies that have been developed capture the economic approach by focusing on specific economic development theories. This article initially provides a theoretical response to the need for a joint approach to the different economic theories relating to Smart Cities, placing the bases of their development in the circular economy. Subsequently, the paper presents a device-based proposal to validate the sustainability principles indicated in the Smart Economy, focusing exclusively on the areas of health and mobility. As a whole, the work concludes with the need to incorporate sustainability criteria into economic ambition so that technological developments have a place in future Smart Cities.

**Keywords:** mobility; smart economy; smart services



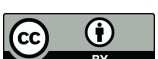

## 1. Introduction

The development of wireless communication technologies that we have been witnessing in recent years has lead to the development of a multitude of interconnected devices capable of providing relevant information about the environment and the users of these devices. This is known as the Internet of Things [1].

One of the fields in which this type of device is being applied largely is in Body Area Network solutions. Due to the conditions of this type of device, the design tends to be miniaturised and, therefore, does not usually have a high-capacity battery. Protocols such as Bluetooth Low Energy, a low-energy extension of the Bluetooth protocol [2], provide wireless connectivity and long battery life for these devices.

The ease of data acquisition and the flexibility of new electronic circuit manufacturing techniques allow us to design devices that can provide users with a better quality of life without being uncomfortable to use [3].

One of the sectors in which this type of device has not yet been widely used is in physiotherapy and rehabilitation treatments [4]. This sector is essential in our society. It helps a person who has suffered some injury return to their normal life, providing them with enhanced quality of life and the ability to return to work at an early stage [5].

This type of treatment traditionally depends on the professional who carries it out. In this way, the patient is monitored according to his or her sensations and the evolution of the process [6]. However, on many occasions, there are periods of unexplainable deterioration. Given the lack of data on the patient's external life, the professional may doubt the efficacy of certain rehabilitation techniques in certain patients.

From an economic point of view, the approach of different technologies has to follow the precepts of participation in the Smart Economy, from the point of view of Smart Services, in such a way that it can be sustainable and thus help to maintain the idea of the Smart City over time.

It is also important to highlight that the development of Smart Cities is based on the contribution made by the theory of the circular economy, since it is undoubtedly based on sustainable developments that we can find sustainability. As can be seen in the previous literature, the circular economy has mainly focused on the study of waste [7–9]. The link between the circular economy and Smart Cities is still under development and a novelty in health and mobility. However, there are certainly studies that have worked on it recently [10].

After this introduction, the article presents in its second section the concept of the Smart Economy, where it also incorporates the necessary connection between Smart Cities and the development of the circular economy. The third section includes a specific case of the application of technological developments to mobility. The fourth and fifth sections correspond to the discussion and conclusions, respectively.

## 2. Smart Economy

The economic development of cities has always stood out for being studied from the perspective of efficiency and effectiveness [11]. One of the most widely studied services in cities is the efficiency of law enforcement agencies [12].

### 2.1. Smart Economy and Smart Cities

The emergence of the concept of the Smart Economy arises from going one step further within the scope of the study of sustainable city development [13]. The move towards cities being and becoming smart is an additional step that is also economically studied by trying to answer questions, such as: does traditional urban economic theory apply to Smart Cities; what do new transport, commerce and communication services consist of, and how do they impact the Smart City economy; or can Smart Cities be inclusive? [14,15].

In the current article, we will focus on the question of new transport and communication services and the ability of new cities to be inclusive.

A key element in the development of Smart Cities is Smart Services. Without Smart Services, it would not be possible to respond to the need to incorporate adequate transport, communication and inclusion [16]. By Smart Services, we mean services that serve the citizens of Smart Cities and their individual needs. These are constantly changing in terms of information and communications technology. In the joint creation of value, the interactions between citizens and service providers take on their full immensity [17].

An essential aspect to highlight concerning the development of mobility solutions relates to the development of the circular economy, which is the basis for the development of Smart Cities [18]. One of the proven connections between the circular economy and the development of Smart Cities through Smart Services is the recycling of urban waste [19,20].

The continuous improvement of the smart service is the basis of the functioning of Smart Services and is the point of connection with the principles of the circular economy. In addition to continuous improvement, Smart Services incorporate the strategic, operational, design and transition cycle [21].

### 2.2. Circular Economy and Smart Cities

The circular economy is the basis of this article, understanding its development as necessary so that these precepts can be incorporated into the different devices and thus contribute to the sustainable development of Smart Cities [22].

An important idea that stands out in the circular economy is the cycle of materials, which is closely related to the concept of industrialisation [23]. We have to start from the idea that the circular economy aims to reduce negative environmental impacts and stimulate new business opportunities within industrialisation, even though the linear flow has dominated overall development causing severe environmental damage, as the diagram could be seen in Figure 1 [24].

The circular economy concept is not entirely new; it has gained a lot of momentum among both scholars and practitioners. Nowadays the concept has become popular, especially due to its promotion by the world's leading powers. It is at this point where this

concept becomes essential in this article, due to its necessary repercussions on the Smart Cities concept, so that all the developments carried out in them are by the sustainability of Smart Cities and Smart Services.

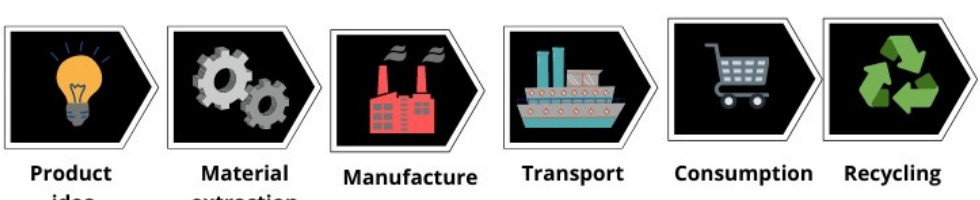

**Figure 1.** Linear economy.

The real importance of the smart urbanism, which is ultimately given by the arrangement of Smart Cities around new technologies, has been implied in several studies, among which it is worth mentioning [25–27]. Adaptation to the new urbanism has been studied before and it is now the different verticals exposed in the current research that we have to deepen, i.e., those related to health and mobility.

What is new and particularly interesting about the circular economy is that it provides an economic system with an alternative flow model to the traditional linear model of extraction, use and disposal of materials, as well as to the energy flow of the modern economic system, which over time has become unsustainable, and has a major impact on the development of Smart Cities [28].

Circular economy as a concept emerged in the late 1970s, attributed to Pearce and Turner [29], who described how natural resources influence the economy by providing inputs for production and consumption. Added to this is the work of Boulding [30], who describes the earth as a circular place with a limited assimilative capacity, so it is not illogical to think that it is necessary and vital that both economy and environment coexist in equilibrium, and in our case at the city level, as the diagram could be seen in Figure 2.

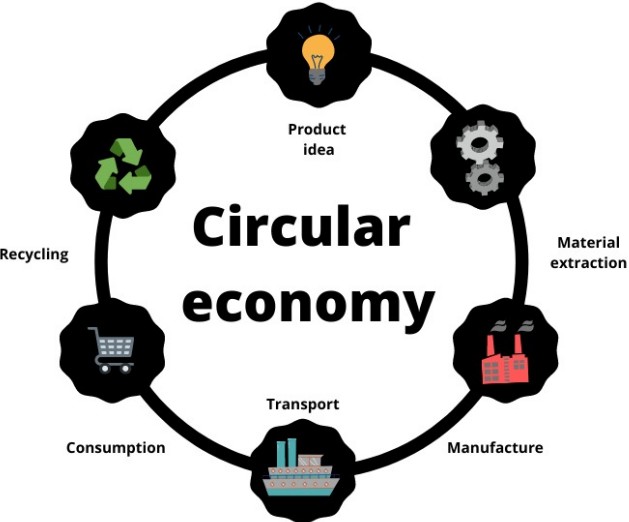

**Figure 2.** Circular economy.

It should be noted that the coupling between the circular economy and Smart Cities is based on the definition of the process as thinking about all aspects of the economy of cities as a whole, as something circular, instead of thinking about it as something linear in such a way that in each phase the following premises are taken into account:

- Continuous thinking about better production, leading to maximum savings in materials.
- Making waste generated in the manufacturing process profitable.
- Minimisation of the extraction of raw materials, saving the high costs faced by organisations.
- Continuous and targeted thinking towards the creation of recycled materials. Plan for the renovation of waste for its reuse and profitability.

In addition to the above, there are necessary approaches to address from the perspective of Smart Cities governance. The application of business in Smart Cities [31] should be focused on reusing products, materials and components, remanufacturing, repair, renovation and improvement. This needs to incorporate solar, wind, biomass and the reuse of energy derived from the waste throughout the product value and life cycle [32].

Another approach necessary for developing the total Smart City concept from an economic point of view is the industrial approach. Appropriate industrial strategies for waste prevention, job creation in the city, efficiency and dematerialisation of the industrial economy have to be described [33]. It is important to incorporate the idea of using rental as a sustainable business model instead of asset ownership for a loop economy, allowing industries and developers to benefit without externalising the costs and risks associated with, for example, waste [34].

Understanding how the circular economy works and its application within Smart Cities as economic systems and industrial processes will allow us to evolve towards Smart Services from the point of view of mobility as well as timely and sustainable technological developments, the bases of the development outlined here.

## 3. Integration of the Circular Economy in the Mobility Aspects of the Smart City

As we anticipated, several studies approximate the importance of sustainability in the development of the Smart Economy, which is key to the development of Smart Cities. Below, within the possible verticals that favour the development of Smart Cities based on sustainability, we have those based on health and mobility as a developed use case.

The device mentioned in this article aims to provide real-time data on the support utilised by patients undergoing mobility enhancement, to be analysed by mobility support staff [35,36].

To this end, this development has been patented under utility model ES1249804U [37]. Concerning the state of the art, the invention incorporates specific improvements, such as a measurement system contained in the rubber tip of the crutch, unlike others where it is located inside the cane to be used in any crutch. In addition, thanks to the functionality of the mobile application, it assists the monitoring of patients with different bone injuries by the specialist, as could be seen in Figure 3. Employing a series of sensors and electronic devices that provide wireless connectivity makes it possible to control, know and correct in real-time the percentage of load carried by a patient undergoing rehabilitation in each support during ambulation.

Thanks to the measurement of anthropometric parameters, we are able to establish estimates of the user's state in real-time, understanding by their state the situation in which a set of determined biometric measurements is within pre-set thresholds [38]. Through the development of a prototype such as the one described, the aim is to improve and speed up the physical and motor recovery of the patient, offering support ranges and generating beneficial feedback for the rehabilitator who, by accessing the data generated by the device and stored in the user's mobile phone, can propose corrections on how to support weight when walking and guide the rehabilitation sessions according to the specific case.

During the design of the device, research was carried out into techniques for integrating a load cell into the tip of the crutch or cane (in the area in contact with the ground), which transforms the pressure variation data into an electrical potential variation capable of

being read from an Analog–Digital Converter HX711. This 24-bit converter is responsible for transforming the analog voltage signal from the load cell into a digital signal and communicating it to the ESP32 control and communications module.

Thus, after processing the data, and thanks to a wireless communications module included in the ESP32 (low-power SoC chips with integrated Wi-Fi and dual-mode Bluetooth technology), it transmits the information to a smartphone via radio, using the Bluetooth Low Energy (BLE) protocol [38]. Once the mobile device obtains the data, either by a mechanical stimulus (vibration) or by an acoustic stimulus (beep), it indicates to the patient if he/she is outside the support range indicated by his/her doctor and/or physiotherapist, thus allowing him/her to self-correct.

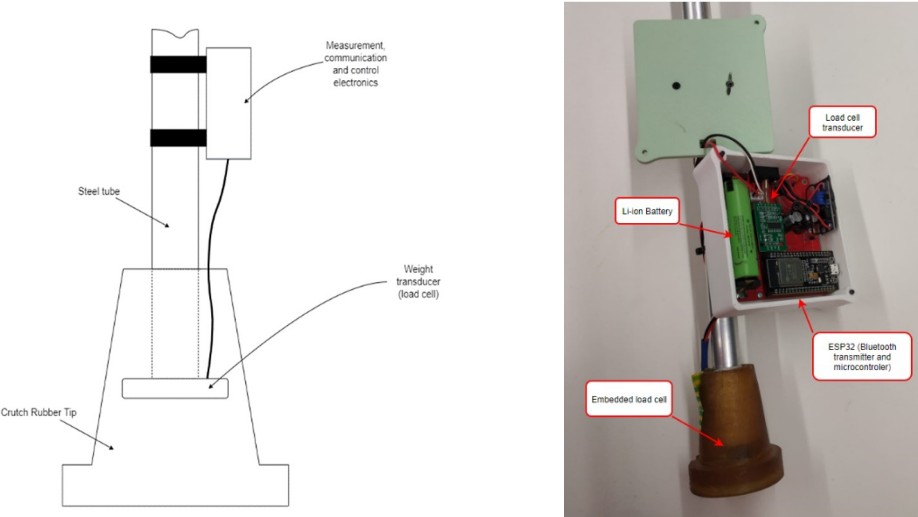

**Figure 3.** Diagram of the device (**left**) and device developed (**right**).

To achieve this purpose, a mobile application has been designed to control, store and visualise the data transmitted by the device. The mobile phone reads the patient's battery percentage and weight data thanks to the connection from the unique UUID. The integrated software sends an average of 30 data in 300 milliseconds. This provides a tool with which both the patient and the rehabilitator can access the history of data collected during the programmed recovery phases. Additionally, the rehabilitator is able to configure the rehabilitation phases and establish the correct support ranges for each of these phases.

To monitor the recovery process, the concept of a phase is used. A phase comprises, in a given period of time, the optimal conditions for the healing process to take place: the number of crutches the user needs to use and the weight to be supported on them. The main screen of the mobile application shows the previous day's activity, as could be seen in Figure 4: the steps recorded, the battery status of the device, the patient's recovery phase and an analysis of the steps recorded. From this screen it is possible to customise the notifications for exceeding the support limits set by the therapist, connect the crutches and consult the application's records.

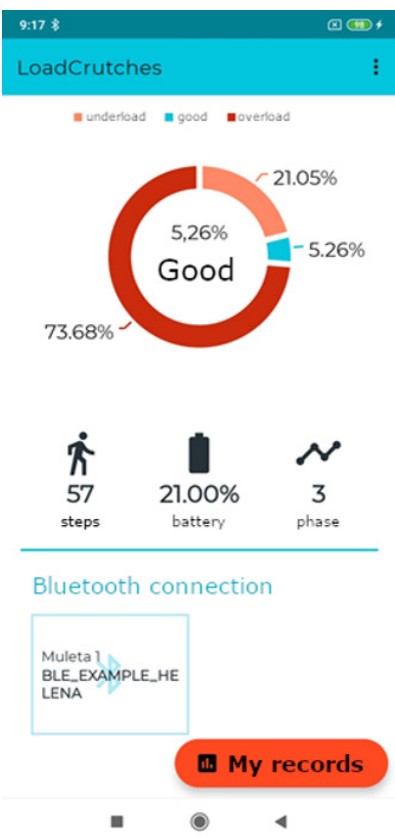

**Figure 4.** Mobile application overview.

## 4. Discussion

The technological development of Smart Cities is widely studied from a technological point of view. From the economic point of view, there are various approaches to the concept of Smart Cities. The economic approaches arise in one direction or another, i.e., on the one hand, we have those who approach the concept of the Smart Economy from the point of view of the incorporation of Smart Services. At the same time, other authors link it to the circular economy.

In the current article, we link a use case based on mobility solutions to the economic sphere of Smart Cities. At the same time, it meets the sustainability criteria of the circular economy, which motivates the so-called Smart Economy.

In the current article, the chosen solution is based on Smart City communication solutions and, more specifically, on mobility-related solutions. In the future, different studies can focus on the globality of the Smart Economy, applying not only to mobility criteria but also to other types of criteria related to other services.

At the same time, the economic verticals presented here are developed on the idea of sustainability linked to the circular economy. However, other ways of coupling to economic theory can be explored.

## 5. Conclusions

This article presents the relationship between technological developments framed within the improvement of Smart Cities and their development within a sustainable framework such as the Smart Economy.

Within the Smart Economy, a link also exists with Smart Services, the latter being those that have to be developed following the precepts of the circular economy and, therefore, those of sustainability. The developments being worked on are those related to mobility, which is the main communication axis of any Smart City. Based on co-communication and mobility as the driving forces of the Smart City philosophy, we advance the relationship they must have with the so-called Smart Economy and, more specifically, with the circular economy.

This article lays the foundations for the necessary work on sustainability from the point of view of the main precepts of the circular economy and the so-called Smart Economy, to make progress in linking them with sustainable technological developments over time.

Theoretically, the link between developments integrated into the Smart Economy and the circular economy idea is presented and conceptualised to succeed over time. As a motivating factor for the need to incorporate the economic reality of Smart Cities into technological developments, new developments must contain the following bases:

Product idea
Material extraction
Manufacture
Transport
Consumption
Recycling
And product idea again

It is important to conclude that if the current scheme is the one followed by any technological development, whether or not it is linked to the idea of mobility, it is theoretically approximated by the current article that its future link with the Smart City ecosystem will have a place within the premises of mobility and sustainability.

**Author Contributions:** Conceptualization, J.P.-D. and F.D.l.P.; methodology, J.P.-D., J.H.S., S.M.-S., A.G.-B. and F.D.l.P.; software, J.H.S., S.M.-S., A.G.-B. and F.D.l.P.; validation, J.P.-D., J.H.S. and F.D.l.P.; formal analysis, J.P.-D. and F.D.l.P.; investigation, J.P.-D., J.H.S., S.M.-S., A.G.-B. and F.D.l.P.; resources, J.P.-D., J.H.S., S.M.-S. and F.D.l.P.; data curation, J.H.S. and S.M.-S.; writing—original draft preparation, J.P.-D. and J.H.S.; writing—review and editing, S.M.-S. and F.D.l.P.; visualization, J.P.-D. and F.D.l.P.; supervision, J.P.-D. and F.D.l.P.; project administration, J.P.-D.; funding acquisition, J.P.-D. All authors have read and agreed to the published version of the manuscript.

**Funding:** This research received no external funding.

**Institutional Review Board Statement:** Not applicable.

**Informed Consent Statement:** Not applicable.

**Acknowledgments:** This work has been partially supported by the European Regional Development Fund (ERDF) through the Interreg Spain–Portugal V-A Program (POCTEP) under grant 0677\_DISRUPTIVE\_2\_E (Intensifying the activity of Digital Innovation Hubs within the PocTep region to boost the development of disruptive and last generation ICTs through cross-border cooperation).

**Conflicts of Interest:** The authors declare no conflict of interest.

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
