# Peer review of "Technological Developments of Mobility in Smart Cities. An Economic Approach"

_smartcities, doi:10.3390/smartcities4030050_

Round 1
Reviewer 1 Report
I recommend to the authors, include reference for the next paragraphs. the circular economy concept is not entirely new; it has gained a lot of momentum 96 among both scholars and practitioners. Nowadays, the concept has become popular, es- 97 pecially linked to its promotion by the world's leading powers. the work is correctly described, the following comments could contribute to the structure: 1) Expand the description of the device components. 2) Describe the type of material for recycling and the contribution in the economic model.Author Response
Dear reviewer, please find attached a file with the answers, kind regards.

Reviewer 2 Report
The general idea of the paper is of particular relevance in the area of smart cities and their relationship with the concept of circular economy.
Nevertheless, the paper needs more insights into more specific scenarios (health-rehab, transportation, communication) and the transversal supporting technologies as mobile phones (and their sensors).
The paper lacks of structure and a continuous story-telling and presentation of more quantitative indicators in which the scenarios - technologies could/should be associated in the circular economy concept.
Author Response
Dear reviewer, please find attached a file with the answers, kind regards.

Reviewer 3 Report
The following are my conclusions:
- The idea is interesting, and several articles in different domains faced the problem.
- The contribution of the paper is not clear. The paper is not well positioned concerning other works in the similar research field.
- The Introduction is very general and not actually introducing the main topic of the paper.
- However, I would say that this paper is just a mere presentation of the problem statement with no description about a real implemented scientific contribution.
- The structure of the paper and technical quality need to be improved.
- It would be useful to enrich the bibliography with some scientific papers.
Author Response

(The authors gave the same response as above.)

Reviewer 4 Report
On the plus side, this paper addresses an interesting and important theme of research, but there are 2 main problems.
First, the structure is not solid and coherent and it is not clear what the authors are seeking to do with this paper. This reads more like a lit review and theoretical paper, but towards the end the authors talk about a physical device that is under development. Adding to the confusion, is the fact that the paper begins by focusing on economic themes, while the device focuses on health and mobility. This empirical part is weak, and I suggest the authors focus on a review of economic themes related to smart cities and keep their contribution theoretical.
Second, there is actually a lot of literature on the economic dimensions of smart cities (particularly from the fields of urban studies, planning and geography) that should be acknowledged and it is missing here. I recommend some relevant urban studies which discuss the economic side of smart cities:
Cugurullo, F. (2021). Frankenstein Urbanism: Eco, Smart and Autonomous Cities, Artificial Intelligence and the End of the City. Routledge.
Angelidou, M. (2015). Smart cities: A conjuncture of four forces. Cities, 47, 95-106.
Zvolska, L., Lehner, M., Voytenko Palgan, Y., Mont, O., & Plepys, A. (2019). Urban sharing in smart cities: the cases of Berlin and London. Local Environment, 24(7), 628-645.
There is also a lot of literature on smart urbanism and neoliberalism that tackles critically the economic side of smart cities.
Author Response

(The authors gave the same response as above.)

Round 2
Reviewer 2 Report
I appreciate the response and the changes authors made in the actual version of the paper.
Nevertheless, the link in section three, presenting the device (lines 162-164) and subsequent lines, are more on the side of rehab than on the mobility use case:
"The device mentioned in this article aims to provide real-time data on the support performed by patients undergoing a rehabilitation process to be analysed by the rehabilitation staff"
It is not clear, how this device is intended to close the loop in circular economy. Please, consider to take the part on mobility, rather than rehab (at least, the explanation of this device as a multi-purpose and the implications in circular economy). This clarification can enhance the paper.
Author Response
Dear reviewer, thanking you for your new considerations, we have incorporated the text again; thank you.
Kind regards,

Reviewer 4 Report
Thanks for addressing my concerns and recommendations. The paper has improved. I still believe that the authors should engage more with the literature on smart urbanism and neoliberal urbanism but overall I am happy these revisions,
Author Response
Dear reviewer, thank you for your considerations.
Kind regards,
